# Effect of Aging Treatment on the Microstructure and Properties of 2.2 GPa Tungsten-Containing Maraging Steel

**DOI:** 10.3390/ma16144918

**Published:** 2023-07-10

**Authors:** Shun Han, Xinyang Li, Yu Liu, Ruming Geng, Simin Lei, Yong Li, Chunxu Wang

**Affiliations:** Research Institute of Special Steels, Central Iron and Steel Research Institute, Beijing 100081, China; hanshunfa@126.com (S.H.); lixinyang@nercast.com (X.L.); liuyu@nercast.com (Y.L.); gengruming@nercast.com (R.G.); leisimin@nercast.com (S.L.); liysea@163.com (Y.L.)

**Keywords:** maraging steel, aging treatment, reverted austenite, martensite lath, precipitations

## Abstract

Maraging steel is a prominent category of ultrahigh-strength steel (UHSS) characterized by excellent comprehensive properties, and it finds wide applications in manufacturing load-bearing structural components. In this study, a novel tungsten-containing maraging steel, C-250W, was designed. The effects of aging treatments on the mechanical properties, microstructure, precipitations, and reverted austenite of C-250W steel were investigated. The results revealed that the optimal combination of strength and toughness could be achieved through an aging treatment of C-250W steel carried out for 5 h at 480 °C after solution treatment at 1000 °C for 1 h. As the aging temperature increased, the proportion of dimples in the impact fracture gradually decreased while that of quasi-cleavage increased, leading to a reduction in Charpy impact energy. The boundary of martensitic lath decomposed gradually as the aging temperature increased, and it disappeared entirely at temperatures higher than 550 °C. Moreover, the aging process resulted in the formation of phases, including spherical Fe_2_M (M represents Mo, W) and thin strip-shaped Ni_3_N (N represents Mo, Ti) precipitates. These precipitates coarsened from 5 nm to 50–200 nm with increasing aging temperature. Additionally, the content of reverted austenite increased with the aging temperature. Within the temperature range of 400 °C to 500 °C for aging treatment, the content of film-shaped reverted austenite was approximately 3%, primarily distributed at the boundary of martensite lath. When the aging temperature exceeded 550 °C, the content of reverted austenite reached 20.2%, and its morphology changed from film-shaped to block-shaped, resulting in a decline in strength and toughness.

## 1. Introduction

Maraging steel is a prominent category of ultrahigh-strength steel (UHSS) characterized by a yield strength exceeding 1350 MPa and excellent comprehensive properties [1,2]. It finds wide applications in manufacturing load-bearing structural components for aerospace, navigation, energy, and other fields [3,4]. Different from traditional high-strength steel that relies on carbide strengthening, maraging steel consists of a Fe-Ni martensitic matrix in which intermetallic compounds precipitate in the supersaturated solid solution (martensite) diffusely to achieve strengthening [5,6,7].

In the 1960s, Bieber and Decker [8] significantly improved the age-hardening ability of maraging steel by adding Co and Mo elements to the Fe-Ni alloy, leading to the development of the 18Ni250 (also known as C-250) with a yield strength of approximately 1700 MPa. Owing to the scarcity and the rising cost of cobalt as a strategic resource globally, the production cost of maraging steel has significantly increased, posing limitations on its further development and application. Therefore, researchers have aimed to develop cobalt-free maraging steels with properties comparable to those of traditional maraging steel. International Nickel (INCO) and Teledyne Vasco jointly developed T250 maraging steel with a yield strength of 1750 MPa, where T represents the Ti-strengthened type [9]. T-250 maraging steel achieves the strengthening effect by dispersing and precipitating nano-scale intermetallic compounds on the soft Fe-Ni matrix with ultra-low carbon content. Cobalt-free T-250 maraging steel, with reduced Mo content and Ti-doped strengthening phases, performs similarly to C-250 maraging steel. Kim and Lee et al. [10] developed cobalt-free maraging steel W-250 with tungsten (W) replacing Mo as the strengthening element, and the precipitates formed were Ni_3_W and η-Ni_3_Ti. Tharian and Sinha [11] developed a low-Ni and low-Cr Co-free maraging steel, significantly reducing the production cost. Ren et al. [12] previously developed copper-containing maraging alloys, while He et al. [13] designed cobalt-free maraging steel with properties similar to T300, exhibiting a strength of 2080 MPa. However, further development is needed to balance strength and toughness. Table 1 presents the chemical composition and optimal mechanical properties of maraging steels developed based on the typical C-250 steel.

For maraging steel, the precipitation behavior of precipitates greatly affects the strength of steel. Tewari et al. [17] investigated the precipitation behavior of Ni3(Ti,Mo) in 18 Ni maraging steel; the results indicated that the evolution of particles changed from the collapse of the unstable BCC lattice to clustering and ordering of atomic species with increasing aging temperature, of which the critical temperature was 450 °C. Jiang et al. [18] reported a novel maraging steel via minimized lattice misfit of nanoprecipitation, controlling the precipitates of Ni(Al,Fe) to enhance strength and obtaining a strength–ductility combination of 2.2 GPa-8.2%. A great strength increment of about 1100 MPa was obtained with nearly no reduction in ductility in the aging treatment.

Moreover, the formation of film-shaped reverted austenite during aging greatly affects the plasticity of maraging steel. Carvalho et al. [19] indicated that the reverted austenite was formed at interface regions, including boundaries of grains, packages, and martensite laths, at 520 °C, which was further observed in the interior of martensite laths above 560 °C. Reverted austenite with Ni-riched not only improved the plasticity of the maraging steel [20] but also improved cryogenic plasticity [21,22] and fatigue performance [23,24].

In this study, tungsten-containing maraging steel, C-250W, was designed based on the typical composition of C-250. In order to clarify the strengthening and toughening mechanisms of the steel and obtain optimal mechanical properties, the effect of aging treatment on the microstructure and properties of tungsten-containing maraging steel was studied in the present study. Mechanical performance testing and microstructure morphology observation were conducted to analyze changes in mechanical properties, precipitation, and reverted austenite during the aging process.

## 2. Material and Methods

### 2.1. Test Material 

The experimental alloy containing tungsten was named C-250W steel, and its chemical composition is presented in Table 2. The C-250W steel was smelted in a vacuum induction furnace using ultra-high-purity pure iron and metal materials such as Ni, Mo, Co, Ti, Al, and W. The steel ingots were homogenized at 1200 °C and then forged into billets with various specifications. During the forging process, the initial forging temperature was 1180 °C, and the final forging temperature was maintained above 850 °C. 

First, tensile and impact samples were austenitized for 1 h and then oil-cooled. Subsequently, aging treatments were performed in two patterns involving different temperatures and times. For one group, the samples were subjected to aging treatment for 5 h at temperatures of 400 °C, 430 °C, 460 °C, 480 °C, 500 °C, 520 °C, 550 °C, and 600 °C, respectively, to study the effect of aging temperature on mechanical properties. For the other group, the samples were aged at 480 °C for 3 h, 5 h, 8 h, 10 h, 15 h, and 20 h to investigate the effect of aging time. Finally, the heat-treated samples were machined into standard specimens for tensile strength and Charpy impact. 

### 2.2. Mechanical Testing and Microstructure Characterization

Tensile tests under a stretching rate of 1 × 10^−2^ m/min were conducted at 25 °C using a universal material testing machine (MTS-880, MTS Corporation, Woodbury, Minnesota, USA), in which the samples were machined to a shape with a parallel section diameter of 5 mm and a length of 65 mm, as shown in Figure 1. Charpy “U” notch impact tests were performed at room temperature on an impact testing machine (Ni300, NCS Testing Technology Corporation, Beijing, China) using samples with dimensions of 10 mm × 10 mm × 55 mm. Rockwell hardness (HRC) measurements were carried out using a hardness testing machine (TH300, TIME Corporation, Beijing, China) with a load of 150 kg. The samples were etched using a mixture solution of 5 g CuCl_2_ + 30 mL HCl + 30 mL H_2_O + 25 mL alcohol to observe the austenite grains and microstructure under a metallographic optical microscope (OM, GX51, Olympus Corporation, Tokyo, Japan). For the volume fraction of austenite measurement, polished samples were etched with a mixture corrosion solution of 1% tetramethylammonium chloride + 10% acetylacetone + methanol, and an X-ray diffractometer (XRD, Philips APD-10, Philips Corporation, Amsterdam, The Netherlands) was utilized. Samples after impact tests were prepared to observe fracture morphology using a cold field emission scanning electron microscope (HITACHI-S4300, Hitachi Corporation, Tokyo, Japan). Polished samples taken from impact samples were etched with a 4% natal solution to observe the microstructure and undissolved precipitations using SEM, and the compositions of the undissolved precipitations were determined using the equipped EDS (EDAX Genesis 6.0, EDAX Inc., Philadelphia, PA, USA). Thin transmission electron microscopy (TEM) foils were prepared from 3 mm diameter discs and ground to 50 μm thickness, then electro-polished using a mixture solution of 10 vol.% HClO_4_ + 90 vol.% C_2_H_5_OH at −20 °C. Further detailed analysis of microstructures was conducted using TEM (Hitachi H-800, Hitachi Corporation, Tokyo, Japan) with an acceleration voltage of 200 kV.

## 3. Results and Discussion

### 3.1. Thermodynamic Analysis

To investigate the phase transformation during the cooling process, the equilibrium precipitation process within the temperature range of 400 °C to 1200 °C was calculated using Thermo-Calc 2022 software (Thermo-Calc AB Corporation, Stockholm, Sweden) with the TCFE12 database. Figure 2 illustrates that the alloy completely transforms into an austenite structure at 980 °C. Based on the thermodynamic calculations, it was determined that the solid solution temperature needed to be higher than 980 °C to achieve the complete austenitization of C-250W steel and dissolve the precipitates. Therefore, a solid solution temperature of 1000 °C was selected for achieving complete austenitization.

### 3.2. Effect of Aging Treatment on Mechanical Properties of C-250W Steel

The C-250W steel underwent solid solution treatment at 1000 °C for 1 h followed by oil cooling, and subsequently, aging treatment was performed at temperatures ranging from 400 °C to 600 °C for 5 h. The impact of aging temperature on the mechanical properties of C-250W steel is illustrated in Figure 3. As the aging temperature increased, the tensile strength (R_m_), yield strength (Rp_0.2_), and Rockwell hardness (HRC) exhibited a similar trend. The precipitation phases increased in size but did not fully grow at the aging temperature of 480 °C, remaining in a semi-coherent state with the matrix. When the aging temperature was raised to 500 °C, the yield strength, ultimate tensile strength, and Rockwell hardness reached their peak values of 2189 MPa, 2267 MPa, and 56.4, respectively. However, further increases in temperature led to over-aging of the C-250W steel, resulting in a decline in strength and hardness. This can be attributed to the coarsening of precipitates, which disrupts their coherent relationship with the matrix. 

With increasing aging temperatures, the elongation (A) and reduction of area (Z) of C-250W steel initially decreased below 500 °C and then began to increase. The mathematical description of the elongation and reduction can be expressed as the following equations:A = (Lo − Lu)/Lo × 100%(1)
Z = (So − Su)/So × 100% (2)
where Lo represents the original length of the parallel section, Lu represents the length of the parallel section after fracture, So represents the original area of the parallel section, and Su represents the area of the parallel section after fracture.

At the aging temperature of 500 °C, both A and Z reached their minimum values of 6.5% and 30%, respectively. The impact energy (A_KU_), which represents the toughness of C-250W steel, followed a similar trend to elongation. At an aging temperature of 480 °C, the impact value was 30 J. However, after aging for 5 h at 500 °C, the A_KU_ dropped to only 21 J. The A_KU_ value slightly varied within the temperature range of 500 °C to 550 °C. Considering the overall combination of strength and toughness through aging treatment within the temperature range of 400 °C to 600 °C, the optimal temperature was determined to be 480 °C. At this temperature, the tensile strength (R_m_), yield strength (R_p0.2_), and impact energy (A_KU_) were measured to be 2201 MPa, 2129 MPa, and 30 J, respectively.

After the C-250W steel underwent solid solution treatment at 1000 °C for 1 h and subsequent oil cooling, it was subjected to aging treatment at 480 °C for varying durations of 3 h, 5 h, 8 h, 10 h, 15 h, and 20 h. This investigation aimed to assess the influence of aging time on the mechanical properties of C-250W steel. The effects of aging time on tensile strength (R_m_), yield strength (Rp_0.2_), elongation (A), reduction of area (Z), Rockwell hardness (HRC), and impact energy (A_KU_) are presented in Figure 4. As the aging time increases from 3 h to 10 h, both R_m_ and Rp_0.2_ steadily increase. Between the aging times of 3 h and 10 h, the ultimate tensile strength experiences continuous growth. However, for aging times longer than 10 h, the ultimate tensile strength only slightly increased from 2149 MPa to 2280 MPa. This limited increase can be attributed to the growth and coarsening of the precipitated phase. The maximum values of elongation (A) and reduction of area (Z) were achieved at an aging time of 10 h, where A reached 8.0% and Z reached 41%. Subsequently, both A and Z decreased as the aging process was further prolonged. In Figure 3d, the impact energy (A_KU_) consistently decreased from 36 J to 19 J as the aging time increased. This reduction indicates a decline in toughness with increasing aging. Based on a comprehensive analysis of the aging treatment parameters, it is determined that the optimal aging temperature and time for achieving an optimized balance of strength and toughness in C-250W steel were 480 °C and 5 h, respectively.

### 3.3. Effect of Aging Temperature on Impact Fracture Morphology and Microstructure of C-250W Steel

The impact fracture cracks growth area microstructure of C-250W steel was observed using a scanning electron microscope (SEM) after aging treatment at various temperatures, as depicted in Figure 5. Within the aging temperature range of 400 °C to 600 °C, the impact fracture morphology of C-250W steel predominantly consists of dimples and quasi-cleavages. As the aging temperature increases, the proportion of dimples gradually decreases, while the proportion of quasi-cleavages increases correspondingly.

At an aging temperature of 400 °C, the macroscopic fracture in the impact sample’s expansion zone exhibited plastic deformation (Figure 5a). The microstructure morphology was primarily characterized by dimples with occasional quasi-cleavage, displaying irregular dimple sizes and a prominent tearing edge. As the aging temperature increased, the dimples became smaller and shallower, reducing the area occupied by dimples.

When the aging temperature reached 500 °C, the yield strength and ultimate tensile strength of C-250W steel peaked, while the impact value was lower than aging at 400 °C. The impact fracture morphology mainly exhibited quasi-cleavage, with only a minimal presence of dimples (Figure 5d). At this temperature, crystallization was observed on the surface, leading to minimal macroscopic plastic deformation and a greater number of tearing edges in the microscopic morphology.

As the aging temperature further increased to 550 °C, the impact energy reached its lowest value. The fracture surface displayed limited macroscopic plastic deformation, and more tearing edges were observed. Furthermore, at this temperature, intergranular fracture morphology became evident.

Finally, at an aging temperature of 600 °C, the fracture morphology appeared flat and smooth, with intergranular fracture morphology becoming more prominent.

After aging treatment for 5 h at temperatures ranging from 400 °C to 600 °C, the morphology of lath martensite in C-250W steel was characterized using TEM. Following the solid solution process at 1000 °C for 1 h, martensitic laths with various sizes and orientations were obtained, and these laths experienced recovery and decomposition during aging treatment.

In the aging temperature range of 400 °C to 440 °C, as depicted in Figure 6a,b, the martensite laths appeared parallel and slender, with straight and distinct lath boundaries. At these temperatures, the diffusion rate of atoms was sluggish, resulting in a limited number of dislocation recoveries. As the aging temperature increased, the martensitic laths maintained their bundled shape, and the diffusion of atoms facilitated the rapid recovery of dislocations. Figure 6d,f demonstrate the progressively blurred lath boundaries due to the increasing decomposition of the martensitic microstructure. When the aging temperature reached 550 °C and 600 °C, the lath bundles became nearly unrecognizable, and the boundaries of the martensitic laths disappeared (Figure 6g,h).

### 3.4. Analysis of the Precipitation during Aging Treatment

Following the aging treatment, the precipitations in maraging steel comprised intermetallic compounds such as Ni_3_Mo and Ni_3_Ti, as well as dispersed reverted austenite [25,26]. These intermetallic compounds, Ni_3_Mo and Ni_3_Ti, were thermodynamically metastable phases. During the aging treatment process, the nucleation of precipitated phases occurred, and they enlarged and dispersed within the martensite laths as the temperature increased. Initially, when Ni_3_Mo, Ni_3_Ti, and reverted austenite precipitated, they were coherent with the matrix, forming a staggered lattice structure. The precipitation of reverted austenite helps mitigate the loss of toughness.

The morphology of precipitation in C-250W steel was characterized by TEM at various aging temperatures, as shown in Figure 7. After aging at 460 °C for 5 h, the precipitate phases appeared as dispersed dots within the martensite. These precipitates were uniformly distributed and closely arranged in the martensitic matrix. The precipitated phases were identified through diffraction analysis as Ni_3_Ti and Fe_2_M (M represents Mo or W). The nano-precipitated intermetallic compounds primarily nucleate and grow at lath or austenite grain boundaries. These precipitations strongly impede the movement of dislocations, leading to a rapid increase in the hardness and strength of C-250W steel. At an aging temperature of 480 °C (Figure 7b), the precipitated intermetallic compounds exhibited distinct enlargement and aggregation, forming rod and needle shapes with a length of approximately 5 nm. They were evenly distributed and uniformly arranged within the matrix. Diffraction analysis confirms that the precipitated phases were Ni_3_M (M represents Mo or Ti). Among the precipitated phases, the Ni_3_Mo phase exhibited a higher nucleation rate within the matrix. The nano alloy compounds precipitated in situ owing to the short-range diffusion of elements, forming a coherent interface with the matrix. Additionally, during the aging treatment, the supersaturated Ti and Ni within the matrix formed Ni_3_Ti owing to their short diffusion paths and rapid diffusion rates.

After aging at 520 °C for 5 h, the TEM analysis reveals the morphology and diffraction pattern of the needle-shaped precipitates within the martensite matrix (Figure 7c). The precipitates were orderly distributed and closely packed, significantly larger than those formed at 480° C. Through diffraction calibration, it is determined that the precipitated phase is Ni_3_M (M represents Mo or Ti). The TEM analysis of the phases precipitated during over-aging treatment at 550 °C is shown in Figure 7d. The precipitates coarsened, with the formation of numerous rod-shaped precipitates. The spherical precipitates were significantly enlarged, often connecting or bonding with each other. Due to the growth of these precipitates, the strengthening effect provided by precipitates is weakened.

Diffraction analysis confirms that the spherical and rod-shaped precipitated phases were Fe_2_M (M represents Mo or W) and Ni_3_N (N represents Mo or Ti), respectively. The maximum length of the rod-shaped precipitates was nearly 200 nm, while the maximum diameter of the spherical precipitates was approximately 50 nm.

### 3.5. Generation and Growth of the Reverted Austenite

The TEM observation reveals that after the solution treatment, the matrix structures of C-250W steel consist of martensite and retained austenite. Following the aging treatment at specific temperatures, certain portions of the martensite undergo a reverse transformation into austenite. In quenched alloy steels, retained austenite is minimal, appearing as thin films distributed at the martensitic lath boundaries. These films exhibit weak diffraction peaks in the X-ray diffraction spectrum. The quantity, distribution, and morphology of austenite play a crucial role in determining the toughness of C-250W steel. After the aging treatment, the microstructure of C-250W steel comprised lath martensite and austenite. As the aging temperature increased, the martensite gradually transformed into reverted austenite. The reverted austenite emerged along the boundaries of martensite laths and austenite, facilitated by the diffusion of the austenitizing element, Ni.

The kinetics of reverted austenite can be described by the formula $\xi =1-\exp^{\left(-kt^{1/2}\right)}$, where ξ represents the content of reverted austenite, *t* is the aging time, and *k* is the diffusion rate constant.

The relationship between the volume fraction of reverted austenite and aging temperature is illustrated in Figure 8. It can be observed that below 480 °C, the content of reverted austenite remains relatively constant at approximately 3%. As the aging temperature increases to 520 °C, the volume fraction of reversed austenite rises to 5.9%. Typically, the austenite reverse transformation temperature for maraging steel ranges from 520 °C to 550 °C. When the aging temperature surpasses this range, a significant amount of reverted austenite forms at the grain boundaries of martensitic laths and austenite. In the over-aging state at 550 °C, the increased diffusion rate of metal atoms results in a rapid rise in austenite content to 20.2%, which is detrimental to the strength and toughness of the material.

The morphology of reverted austenite in C-250W steel at different aging temperatures was observed using TEM. It can be observed that the reverted austenite primarily distributes along the boundaries of the martensite laths. At aging temperatures of 430 °C and 460 °C, as depicted in Figure 9a,b, the reversed austenite appeared as thin strips and films at the lath boundaries. This film-shaped reverted austenite exhibited ductility, which helps to alleviate stress concentration at the crack tip and mitigate crack propagation. The presence of this ductile phase can passivate the crack tip, inhibiting crack initiation and impeding crack propagation.

As the aging temperature increased to 480 °C and 520 °C, the film-shaped reversed austenite grew along the lath boundaries, increasing the content of reversed austenite. When the aging temperature exceeded the austenite reverse transformation temperature (As) at 550 °C, the driving force for austenite decomposition became significantly higher than that for martensite decomposition, resulting in earlier martensite decomposition during the aging treatment.

The transformation of film-shaped reversed austenite into massive austenite with increasing aging temperatures had a detrimental effect on impact toughness. The reverse transformation of austenite involves nucleation and growth, with reversed austenite preferentially nucleating at the boundaries of martensite laths or original austenite grain boundaries. As the aging temperature increased, the reverse-transformed austenite coarsened, and adjacent reverted austenite regions at the boundaries coalesced by engulfing the martensite laths, ultimately forming massive austenite.

## 4. Conclusions

In this study, a W-containing maraging steel, C-250W, was developed, and the effects of aging treatment on its mechanical properties, microstructure morphology, metallic precipitates, aging precipitates, and reverted austenite were investigated to determine the optimal aging temperature and aging time. The following conclusions can be drawn:(1)In the aging temperature range of 400 °C to 600 °C, the tensile strength, yield strength, and Rockwell hardness initially increase and then decrease from their peak values of 2189 MPa, 2267 MPa, and 56.4 MPa, respectively. Conversely, elongation, area reduction, and impact energy exhibit the opposite trend. For the studied steel, the optimal balance between strength and toughness for C-250W steel can be achieved through aging treatment at 480 °C for 5 h.(2)The boundaries of martensite laths remain straight when the aging treatment temperature is below 480 °C. However, as the aging temperature increases, reverse decomposition of martensite occurs. When the temperature exceeds 550 °C, the decomposition of martensite becomes significant, leading to the disappearance of lath boundaries.(3)During the aging treatment, the precipitated phases consist of spherical Fe_2_M (M is Mo, W) and stripe-shaped Ni_3_N (N is Mo, Ti). These precipitated phases coarsen as the aging temperature increases.(4)In the temperature range of 400 °C to 500 °C, the content of reverted austenite remains around 3%, and it is mainly observed as thin film-shaped reverted austenite at the boundaries of martensitic laths. However, as the aging temperature reaches 550 °C, the content of reverted austenite increases to 20.2%, and the film-shaped reverted austenite coalesces into massive austenite, which significantly impacts the toughness of C-250W steel.

## Figures and Tables

**Figure 1 materials-16-04918-f001:**
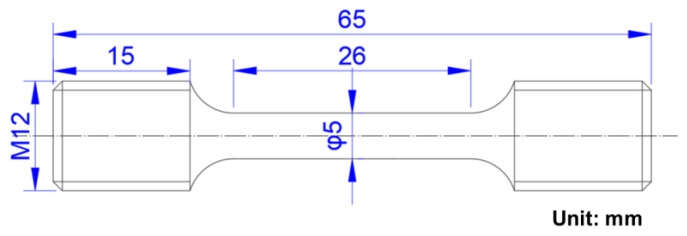
Geometry of tensile test samples.

**Figure 2 materials-16-04918-f002:**
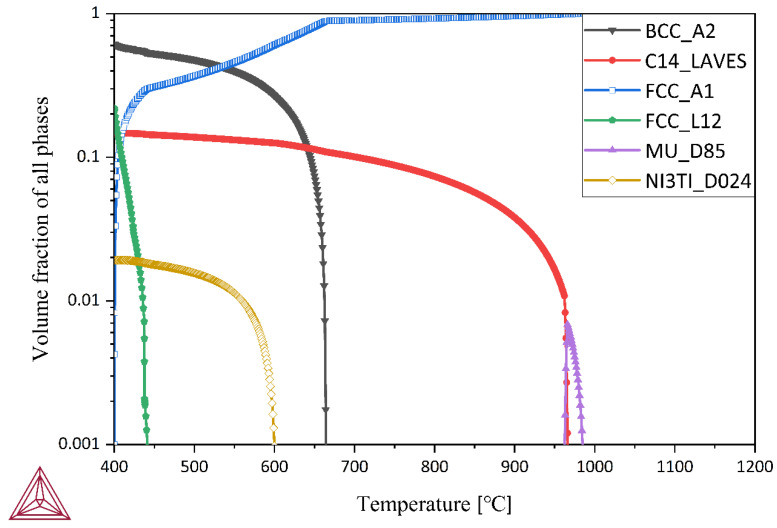
The effect of temperature on phases in the C-250W steel.

**Figure 3 materials-16-04918-f003:**
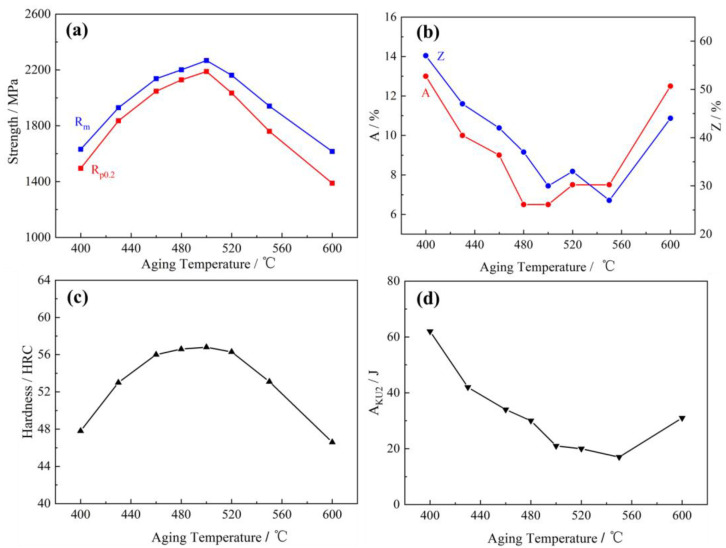
The effect of aging temperature on the mechanical properties. (**a**) Yield strength and ultimate tensile strength, (**b**) Elongation and section shrinkage. (**c**) Rockwell hardness, and (**d**) U-type Charpy impact energy.

**Figure 4 materials-16-04918-f004:**
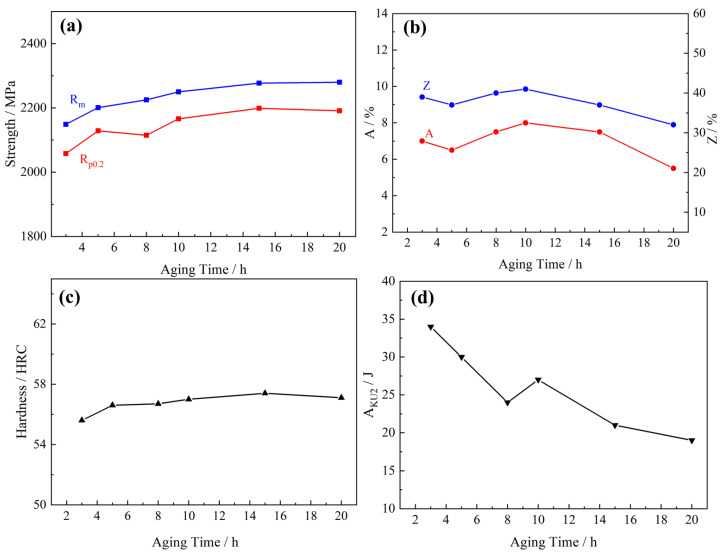
The effect of aging time on the mechanical properties. (**a**) Yield strength and ultimate tensile strength, (**b**) Elongation and section shrinkage. (**c**) Hardness, and (**d**) U-type Charpy impact energy.

**Figure 5 materials-16-04918-f005:**
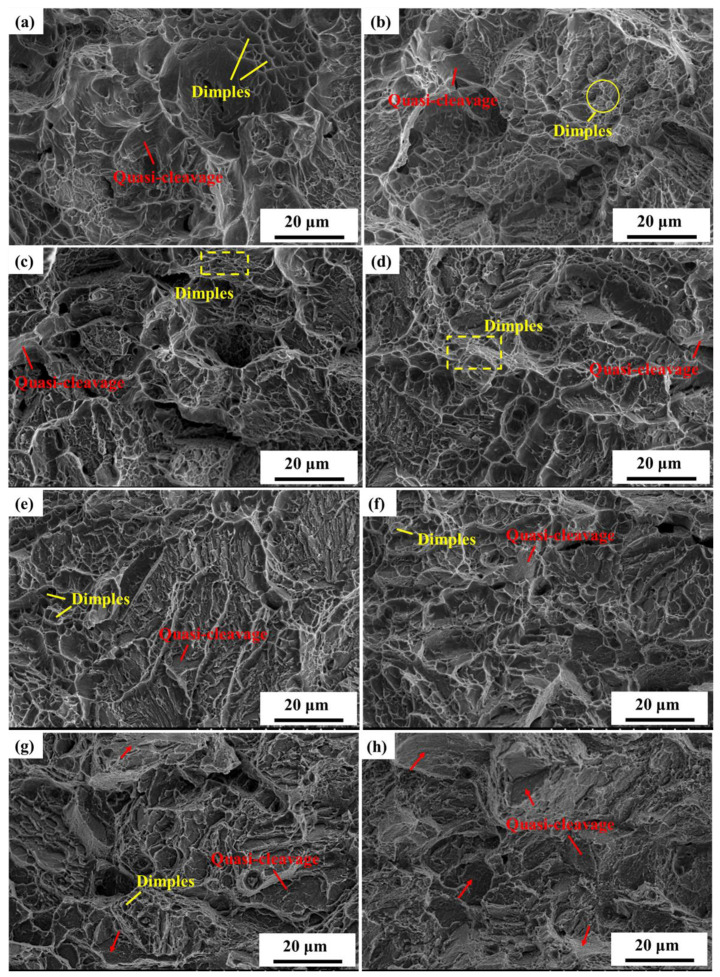
SEM fractography of impact fracture at different aging temperatures. (**a**) 400 °C, (**b**) 430 °C, (**c**) 460 °C, (**d**) 480 °C, (**e**) 500 °C, (**f**) 520 °C, (**g**) 550 °C, and (**h**) 600 °C.

**Figure 6 materials-16-04918-f006:**
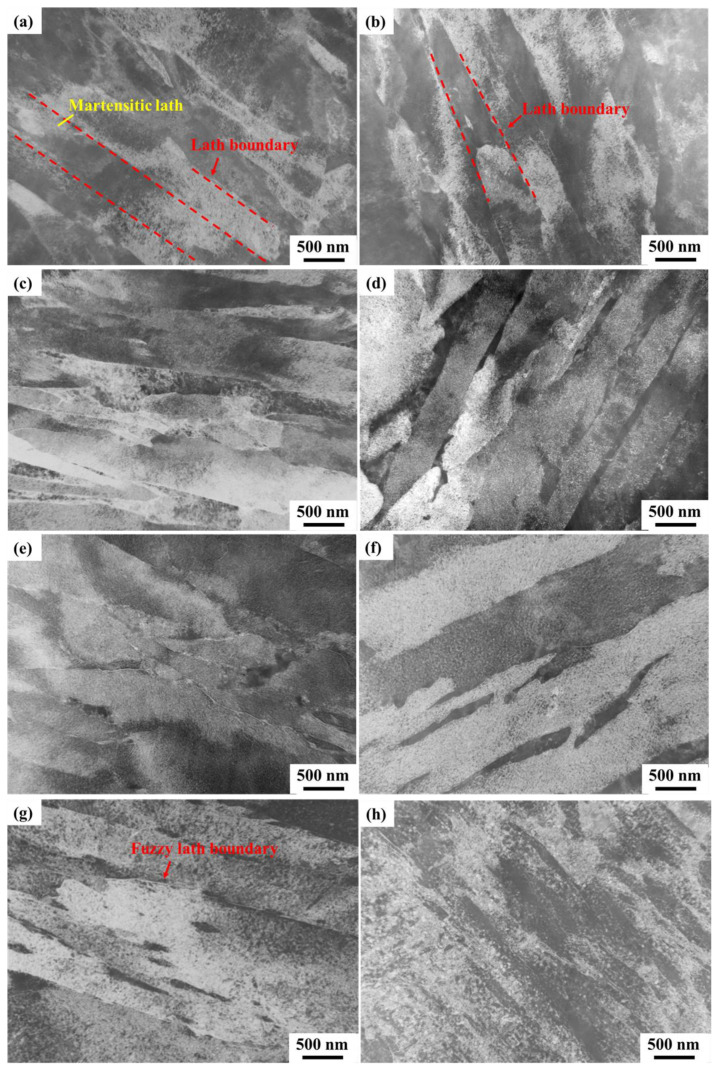
TEM micrographs of lath martensite after aging at different temperatures. (**a**) 400 °C, (**b**) 430 °C, (**c**) 460 °C, (**d**) 480 °C, (**e**) 500 °C, (**f**) 520 °C, (**g**) 550 °C, and (**h**) 600 °C.

**Figure 7 materials-16-04918-f007:**
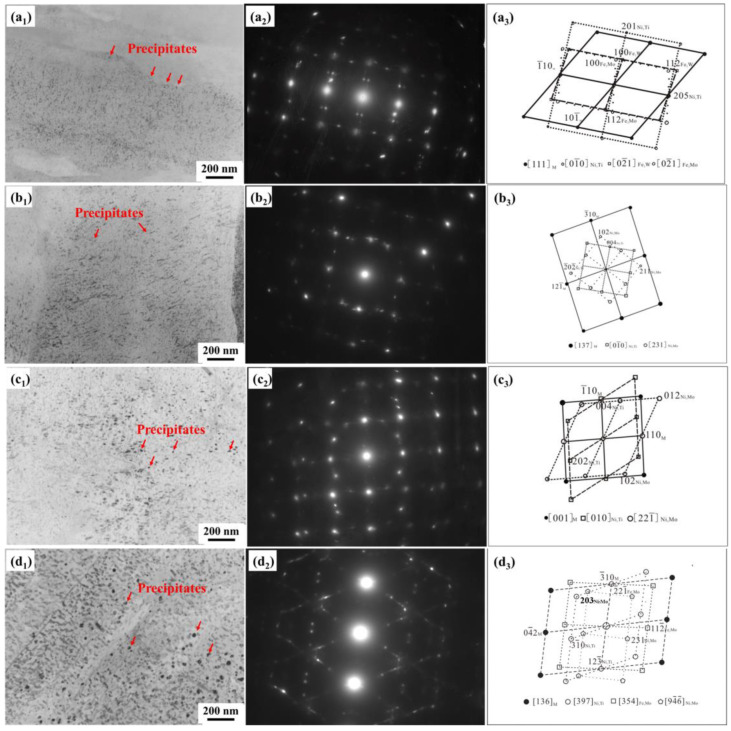
Precipitated phases after aging treatment. (**a_1_**) TEM image of aging treatment for 5 h at 460 °C, (**a_2_**) diffraction spots of (**a_1_**), (**a_3_**) calibration result of (**a_2_**); (**b_1_**) TEM image of aging treatment for 5 h at 480 °C, (**b_2_**) diffraction spots of (**b_1_**), (**b_3_**) calibration result of (**b_2_**); (**c_1_**) TEM image of aging treatment for 5 h at 520 °C, (**c_2_**) diffraction spots of (**c_1_**), (**c_3_**) calibration result of (**c_2_**); (**d_1_**) TEM image of aging treatment for 5 h at 550 °C, (**d_2_**) diffraction spots of (**d_1_**), (**d_3_**) calibration result of (**d_2_**).

**Figure 8 materials-16-04918-f008:**
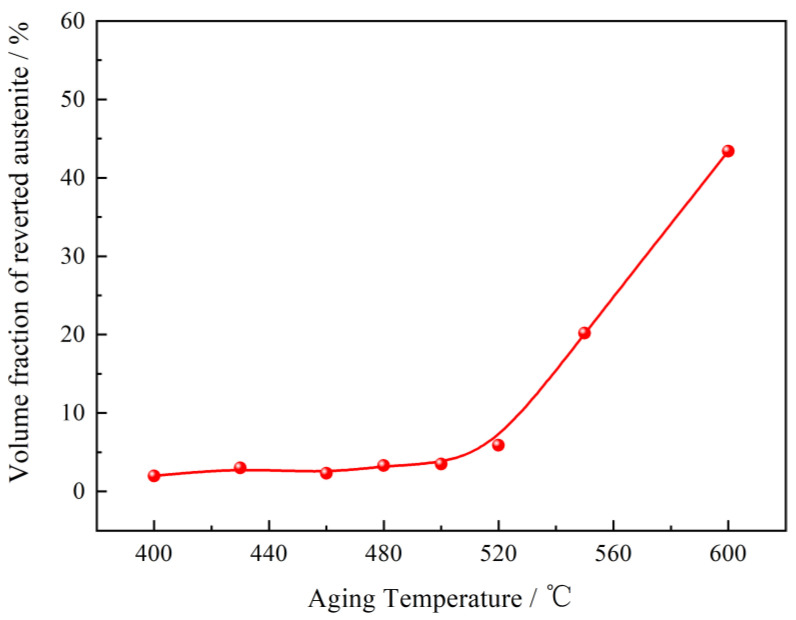
The effect of aging temperature on the volume of austenite.

**Figure 9 materials-16-04918-f009:**
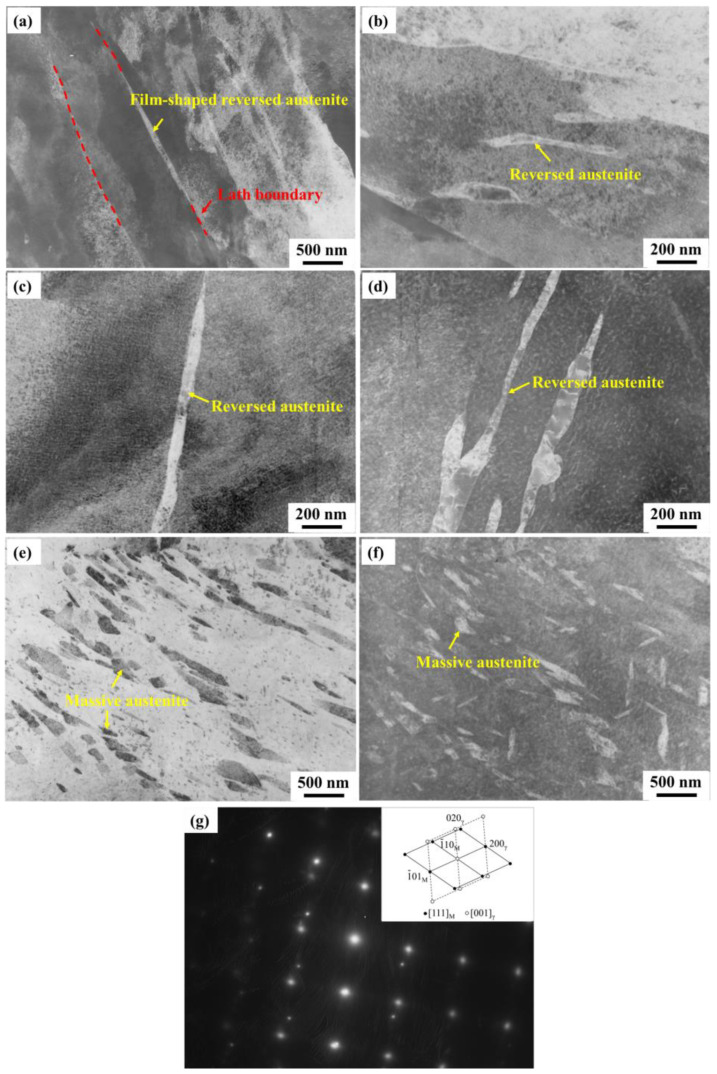
TEM micrographs of reverted austenite after aging at different temperatures. (**a**) 430 °C, (**b**) 460 °C, (**c**) 480 °C, (**d**) 520 °C, (**e**) 550 °C light field, (**f**) 550 °C dark field, (**g**) diffraction spots and calibration of (**a**).

**Table 1 materials-16-04918-t001:** Chemical compositions and mechanical properties of maraging steels [10,11,13,14,15,16].

Steel Series	Chemical Compositions, wt%					Mechanical Properties				
	Ni	Mo	Ti	Al	Other	R_m_ MPa	R_p0.2_ MPa	A %	Z %	K_IC_/ MPa·m^1/2^
C-250	18.5	5.0	0.4	0.1	8.5Co	1800	1700	8	55	120
T-200	18.5	3.0	0.7	0.1	-	1400	1380	10	60	142
T-250	18.5	3.0	1.4	0.1	-	1815	1775	10.5	56	117
T-300	18.5	4.0	1.8	0.1	-	2100	2050	10	54	76
W-250	18.9	-	1.2	0.1	4.2W	1800	1780	9	45	-
14Ni-3Cr-3Mo-1.5Ti	14.3	3.2	1.5	-	2.9Cr	1820	1750	14	65	130
12Ni-3.2Cr-5.1Mo-1Ti	12	5.1	1.0	0.1	3.2Cr	1700	1660	10	-	102
Fe-15Ni-6Mo-4Cu-1Ti	15	6.0	1.0	-	4Cu	-	1785	10	46	-
Fe-18Ni-4Mo-1.7Ti	18	4.0	1.7	-	-	-	2078	9	-	70
Fe-8Ni-3Mn-5Mo	7.8	4.8	0.5	0.2	-	1545	1486	14	40.2	-

**Table 2 materials-16-04918-t002:** Chemical compositions of C-250W steel (wt%).

C	Ni	Co	Mo	Ti	Al	W	Fe
0.0018	17.46	8.18	5.03	0.41	0.092	5.1	Bal.

## Data Availability

Not applicable.

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
