# Peer review of "Effect of Aging Treatment on the Microstructure and Properties of 2.2 GPa Tungsten-Containing Maraging Steel"

_materials, 2023, doi:10.3390/ma16144918_

Round 1

Reviewer 1 Report

General Comments: The study focuses on the design and characterization of a novel tungsten-containing maraging steel, C-250W. It investigates the effects of aging treatments on the mechanical properties, microstructure, precipitations, and reverted austenite of the steel. The study provides valuable insights into optimizing the aging process to achieve the desired combination of strength and toughness. However, there are major revisions required to improve the clarity, organization, and scientific rigor of the study.

Specific Comments:

  1. Introduction: a. Provide a more comprehensive introduction to maraging steel and its applications in load-bearing structural components. Elaborate on the specific challenges and requirements that this study aims to address. b. Clearly state the objectives and research questions of the study to guide the readers throughout the paper.

  2. Methodology: a. Describe the materials and methods used in greater detail. Include information on the composition of C-250W steel, the solution treatment conditions (e.g., temperature, time), and the aging treatment parameters (e.g., temperature, time). b. Provide details on the characterization techniques employed to evaluate the mechanical properties, microstructure, precipitations, and reverted austenite. Specify the equipment used, the measurement protocols, and the statistical analysis employed.

  3. Results: a. Present the results in a more organized manner. Consider dividing the results section into subsections corresponding to the different aspects studied (e.g., mechanical properties, microstructure, precipitations). b. Provide clear and concise descriptions of the observed results. Quantify the changes in mechanical properties and provide statistical data where applicable. Include representative micrographs to illustrate the microstructural changes. c. Ensure that the findings are properly supported by data, figures, or tables. Include appropriate statistical analysis to validate the significance of the observed trends and differences.

  4. Discussion: a. Interpret the results in a more comprehensive manner. Relate the observed changes in mechanical properties, microstructure, and precipitations to the aging treatments and their effects on the material's performance. b. Discuss the underlying mechanisms behind the observed changes. Reference relevant literature and provide a clear scientific rationale for the findings. c. Address any discrepancies or inconsistencies in the results and propose potential explanations or further investigations.

  5. Conclusion: Summarize the main findings concisely and highlight their significance in the context of the study's objectives. Emphasize the practical implications of the results and suggest potential avenues for future research.

  6. Language and Writing Style: a. Improve the clarity and coherence of the writing. Ensure that sentences are grammatically correct and convey the intended meaning. b. Avoid repetitive phrases and unnecessary repetition of information. Condense the content where possible to maintain a concise and focused presentation. c. Use clear headings and subheadings to guide the reader through the paper.

Overall, addressing these major revisions will significantly enhance the clarity, scientific rigor, and impact of the study.

Reviewer 2 Report

The manuscript shows the effect of aging treatment on the microstructure, mechanical properties, microstructure morphology, metal precipitation of tungsten-containing maraging steel C-250W. The research is well designed but presented not very clear. A good comparative analysis of existing publications concerning the tasks set in the work is performed. The methodological section of the manuscript is presented in sufficient detail. The authors used the modern equipment for test of specimens as well as visualization and assistance in the interpretation of the obtained results. They concluded that the  studied steel, the optimal balance between strength and toughness for steel C-250W can be achieved by aging treatment at 480 °C for 5 hours.

 However, some shortcomings should be corrected to make the manuscript acceptable for publication in Materials.

1.     The authors should explain why the yield strength exceeds the tensile strength for T-200 steel (table 1);

2.     The authors should note that the content of retained austenite also affects the crack resistance characteristics of steels (For example, https://doi.org/10.1007/s11003-019-00263-6);

3.     Please state manufacturer, city and country from where equipment has been sourced. This have to be done for each equipment, software, material and chemical in the paper;

4.  It would be expedient for the authors to clearly separate the content of film-like RA and block-like RA of retained austenite, since they have different effects on the properties of steels.

5.     The font size in Figures 2, 3, 6 and 8g should be increased.

6.     Inscriptions in red in fig. 4, 5 and 8 are not recognized.

The authors presented a valuable scientific work. I recommend publishing this paper after correction of the shortcomings.

Reviewer 3 Report

Table 1 should be moved in methods rather than introduction

Can you describe how was maintained the temperature at 1180 up to 850 during forging process ?

For the aging treatment there was any decision making to use that range of temperature and time ?

Tensile test geometry should be provided ?

Sample preparation for TEM and SEM should be described in details

Figure 2,3, 7 requires standard deviation !

Figure 4 should presents in more details the variation from 400 to 600 because as now no much difference between the condition used

Only 600 degree has a difference in respect to other condition used !

The TEM images are very fuzzy and actually do not shows what you claim !

Where are these “interacted with dislocations” in Figure 6 ? as there you indicate some precipitate actually I feel these are not appropriate TEM images !

Can you describe in details how you have calculated “volume fraction of reverted austenite”

Figure 8 is rather SEM then TEM !

Please provide mathematical description for the “the elongation, area reduction”

na

Reviewer 4 Report

Article is of high quality and can be accepted after some minor corrections.

Figure 5 - Please correct the figure labeling. Letter "g" appears instead of "h".

The references format do not follow the Materials journal standard.

Is the TEM analysis consistent with the Thermocalc results ? If yes please add a sentence in the microstructural analysis section. If not justify why.

Fig. 8(g) shows diffraction spots. From which image and area it has been taken ?

Round 2

Reviewer 3 Report

.

.